# Patient Blood Management in Liver Transplant—A Concise Review

**DOI:** 10.3390/biomedicines11041093

**Published:** 2023-04-04

**Authors:** Angel Augusto Pérez-Calatayud, Axel Hofmann, Antonio Pérez-Ferrer, Carla Escorza-Molina, Bettina Torres-Pérez, Jed Raful Zaccarias-Ezzat, Aczel Sanchez-Cedillo, Victor Manuel Paez-Zayas, Raul Carrillo-Esper, Klaus Görlinger

**Affiliations:** 1Critical Care Division, Hospital General de México Dr. Eduardo Liceaga, Mexico City 06720, Mexico; 2Faculty of Health and Medical Sciences, Discipline of Surgery, The University of Western Australia, Perth 6907, WA, Australia; axel.hofmann@ifpbm.org; 3Institute of Anesthesiology, University of Zurich and University Hospital Zurich, 8057 Zurich, Switzerland; 4Department of Anesthesiology, Infanta Sofia University Hospital, 28700 San Sebastián de los Reyes, Spain; antonioperezferrer@gmail.com; 5Department of Anesthesiology, European University of Madrid, 28702 Madrid, Spain; 6Departmen of Anesthesiology, Hospital General de México Dr. Eduardo Liceaga, Mexico City 06720, Mexico; carlaescorza@gmail.com; 7Department of Anesthesiology, Pediatric Transplant, Centro Medico de Occidente, Instituto Mexicano del Seguro Social, Guadalajara 44329, Mexico; bettina27@hotmail.com; 8Surgery Department, Hospital General de México Dr. Eduardo Liceaga, Mexico City 06720, Mexico; jedson1000@yahoo.com.mx; 9Transplant Department Hospital General de México Dr. Eduardo Liceaga, Mexico City 06720, Mexico; aczel@ciencias.unam.mx; 10Gastroenterology Department Hospital General de México Dr. Eduardo Liceaga, Mexico City 06720, Mexico; victormanuelpzmd@gmail.com; 11Academia Nacional de Medicina de México, Mexico City 06720, Mexico; cmx@revistacomexane.com; 12Department of Anesthesiology and Intensive Care Medicine, University Hospital Essen, University Duisburg-Essen, 45131 Essen, Germany; kgoerlinger@werfen.com; 13TEM Innovations GmbH, 81829 Munich, Germany

**Keywords:** patient blood management, adult and pediatric liver transplant, coagulation, viscoelastic testing

## Abstract

Transfusion of blood products in orthotopic liver transplantation (OLT) significantly increases post-transplant morbidity and mortality and is associated with reduced graft survival. Based on these results, an active effort to prevent and minimize blood transfusion is required. Patient blood management is a revolutionary approach defined as a patient-centered, systematic, evidence-based approach to improve patient outcomes by managing and preserving a patient’s own blood while promoting patient safety and empowerment. This approach is based on three pillars of treatment: (1) detecting and correcting anemia and thrombocytopenia, (2) minimizing iatrogenic blood loss, detecting, and correcting coagulopathy, and (3) harnessing and increasing anemia tolerance. This review emphasizes the importance of the three-pillar nine-field matrix of patient blood management to improve patient outcomes in liver transplant recipients.

## 1. Introduction

Orthotopic liver transplantation (OLT) is the standard of care for patients with non-reversible liver disease. It is a challenging procedure encompassing multidisciplinary and coordinated efforts to achieve the best results. Surgical procedures involve significant vessel manipulation in a complex scenario and impaired coagulation due to several factors, including temperature changes, hemodilution, calcium and acid-base imbalance, and other phenomena that may promote bleeding, often leading to the administration of red blood cells to restore oxygen delivery. Nevertheless, the transfusion of blood products in OLT significantly increases post-transplant morbidity and mortality [1,2,3,4] and is associated with reduced graft and patient survival [5,6]. Based on these results, an active effort is required to prevent and minimize blood transfusions and yellow blood products [7,8].

Patient blood management (PBM) is a revolutionary approach, defined as a patient-centered, systematic, evidence-based approach to improve patient outcomes by managing and preserving a patient’s own blood while promoting patient safety and empowerment [8].

This approach is based on three pillars of treatment: (1) detecting and correcting anemia and thrombocytopenia, (2) minimizing iatrogenic blood loss and detecting and correcting coagulopathy, and (3) harnessing and increasing anemia tolerance [9,10,11,12]. The World Health Organization (WHO) recently published a policy brief regarding the urgent need to implement PBM [13].

This review emphasizes the importance of a three-pillar nine-field matrix for patient blood management in improving patient outcomes in liver transplant recipients.

## 2. Coagulopathy in Liver Disease

Patients with advanced liver disease have a hemostatic profile that typically includes thrombocytopenia and reduced coagulation factors; however, these alterations are counteracted by fibrinolysis inhibition, decreased protein C, and increased endothelial-derived von Willebrand factor (vWF) and factor VIII (FVIII). Therefore, this system is considered rebalanced in a precarious equilibrium vulnerable to extrahepatic variables such as infection, renal impairment, and volume status [14,15]. Thrombocytopenia is attributed to splenic sequestration and decreased thrombopoietin; however, this situation is offset by an increase in vWF. In fact, platelet procoagulant activity is fully preserved in patients with cirrhosis, and it has been demonstrated that thrombin generation in patients with liver disease and thrombocytopenia remained normal down to platelet counts of 60 × 10^9^/dL. In addition, platelet transfusion failed to significantly ameliorate clot firmness in adult patients with platelet counts of 50 × 10^9^/dL, assessed using viscoelastic tests [16,17,18]. Although platelet function defects may be present in vitro, the clinical significance of these defects has been debated [19].

Although there are evident defects in the synthesis of vitamin K-dependent coagulation factors and fibrinogen, the synthesis-derived anticoagulant factors, especially protein C, are also reduced. Elevated endothelial-derived FVIII coupled with low protein C contributes to a hypercoagulable state [20,21,22]. In this setting, PT and APTT suggest defective coagulation. However, these tests are not sensitive to deficiencies of anticoagulants or endothelial-derived procoagulant factors and do not represent the balance seen in LT between the pro-and anticoagulant proteins [23].

The fibrinolytic and antifibrinolytic systems may also be imbalanced in patients with cirrhosis. All liver-dependent factors (such as plasminogen) decrease, but tissue plasminogen activator levels are elevated due to decreased liver clearance. Plasminogen is activated by tissue plasminogen activator (tPA) to form plasmin and fibrin degradation products (FDPs). In some populations, this relationship is responsible for a profibrinolytic state. However, evidence suggests that the plasminogen activator inhibitor (PAI) is augmented in some patients with etiologies, such as cholestatic conditions and nonalcoholic steatosis hepatitis, which makes them prone to hypofibrinolysis and a hypercoagulable state [24,25].

Elevated platelet activation markers, thrombin, fibrin generation, and fibrinolysis are expected in patients with liver diseases. Although elevated levels may indicate defective clearance rather than ongoing activation of platelets, coagulation, and fibrinolysis because the liver clears these proteins [26], elevated plasma levels reflect ongoing low-grade disseminated intravascular coagulation with fibrinolysis activation [27].

Coagulopathy in acute liver failure differs from that in chronic liver failure. In patients with acute liver failure, thrombocytopenia is less common than in patients with cirrhosis; coagulation factors decrease in plasma, and fibrinolysis is particularly inhibited, whereas normal or hyperfibrinolysis is present in cirrhosis [28,29]. This phenomenon explains why thromboembolic events increase with greater liver function decompensation. Thus, patients with compensated cirrhosis have a 1% incidence of portal vein thrombosis, whereas patients with decompensated cirrhosis have an 8–25% incidence [30].

However, hemorrhagic events are common, often explained by reduced platelet count, decreased levels of coagulation factors, and fibrinolysis inhibitors [30]. Bleeding is divided into portal pressure-driven, mucosal, or prolonged puncture wound bleeding caused by premature clot dissolution [14].

In the perioperative context of liver transplantation, this rebalanced hemostasis is quickly vulnerable; therefore, decisions must be made to maintain hemodynamic stability and oxygen delivery to the tissues, avoiding futile overcorrections that promote prothrombotic environments, among other catastrophic complications.

## 3. The Importance of Limiting Transfusion in Liver Transplants (Table 1)

### 3.1. Fresh Frozen Plasma

Plasma units require time for thawing and delays transfusion intervention [31]. FFP is ineffective in increasing thrombin generation in patients with cirrhosis, and there is an association with a high risk of transfusion complications (transfusion-associated circulatory overload, portal hypertension, transfusion-related acute lung injury, and mortality). A large plasma volume is required for clinically significant increments because each plasma unit increases factors by 2% to 3% [32,33,34]. It also increases bleeding risk by elevating the capillary and venous pressures. Due to citrate overload and hypothermia secondary to large-volume transfusion, hypocalcemia may hinder improvement in thrombin generation [35]. Severe acidosis impairs fibrin polymerization and thrombin generation, and lactic acidosis may be worsened by impaired elimination during the anhepatic phase of OLT [36,37,38].

Clinical and scientific evidence for the efficacy of plasma transfusion is thus limited, [33,34] whereas the use of hemostatic components and specific coagulation factor concentrates in conjunction with point-of-care tests (POC) are showing the way forward [39,40,41,42].

### 3.2. Platelets

Prophylactic platelet transfusion does not effectively improve the platelet count [43,44,45,46] or viscoelastic clot firmness [39]. Several cohort studies showed that platelet transfusion during OLT is independently associated with higher early mortality compared with non-exposure [43,44]. Adverse effects of allogeneic platelet transfusion, such as transfusion-related lung injury (TRALI) and acute respiratory distress syndrome (ARDS), may be attributed to cytokines release, inflammation, microparticles found in platelet concentrates, possible pathogen contamination, and ABO incompatibility [39,43,44]. Intraoperative platelet transfusion is associated with reduced 1-year graft survival and 90-day or 1-year overall survival [45]. Platelets increase the risk of mortality from sepsis in patients undergoing OLT. These findings were independent of platelet count prior to platelet transfusion. Platelet transfusion should be reserved for patients with active bleeding during liver transplantation, and compensatory fibrinogen substitution should be considered [46,47].

### 3.3. Red Blood Cells (RBC)

RBC transfusion was an independent predictor of significantly increased adverse long-term outcomes [48,49]. Particularly, the transfusion of fresh RBCs might have a potential negative impact on the survival of liver transplantation patients [50]. RBC transfusion is associated with increased perioperative renal dysfunction, reoperation, and the development of postoperative infections in a dose-dependent manner [51]. The requirement of a moderate number of RBC transfusions is associated with a prolonged hospital stay, and transfusion of more than six RBCs significantly diminishes survival and increases re-transplantation rates [51,52].

**Table 1 biomedicines-11-01093-t001:** Transfusion limitation in liver transplant.

Blood Product	Complications	References
Plasma-Containing Blood Components (platelets, FFP)	Graft loss Reduced post-transplant survival TACO (with increased portal hypertension and bleeding) TRALI	[33,34,35]
Platelets	Most associated with increased morbidity and mortality, TRALI	[43,44,45,46,47]
RBC	Artery thrombosis Early surgical reintervention after ORL Reoperation for hemorrhage Graft loss Postoperative infections Prolonged length of stay Decreased one-year survival TACO (with increased portal hypertension and bleeding) AKI	[48,49,50,51,52]

AKI, acute kidney injury; RBC, red blood cell; FFP, fresh frozen plasma; ORL, orthodromic liver transplant; TACO, transfusion-associated circulatory overload; TRALI, transfusion-related acute lung injury.

### 3.4. The Three-Pillar Matrix of PBM in Liver Transplantation (Figure 1)

#### 3.4.1. Anemia and Thrombocytopenia Management

Anemia in patients with chronic liver disease is common. In total, 75% of patients with hepatitis C treatment have anemia [53]. In addition, the pathology of anemia in severe liver disease includes acute and chronic blood loss into the GI tract, micronutrient deficiency, hemolysis, and the treatment of viral hepatitis [54,55,56]. Anemia in liver disease is multifactorial and complicated by chronic inflammation. Erythropoietin administration helps achieve hemoglobin levels in patients with hepatitis C [53]. Additionally, aplastic anemia or side effects of hepatitis treatment with interferon and ribavirin are seen in patients with hepatitis C [56]. In patients with alcoholic liver disease, alcohol may contribute to anemia by malabsorption, malnutrition, or direct toxic effects [57].

Lichtenegger [58] showed an association between preoperative anemia (World Health Organization classification) and survival and complications after OLT. Preoperative anemia was not associated with the one-year survival of patients after liver transplantation [485/599 (81%) OR (95%CI) 1.04 (0.64–1.68), *p* = 0.88]. However, they found higher rates of intra-operative blood transfusions, acute postoperative kidney injury, and postoperative renal replacement therapy. Other authors identified preoperative anemia (Hb ≤ 10 g/dL) as a predictor of massive transfusion and one-year patient and graft survival [59].

**Figure 1 biomedicines-11-01093-f001:**
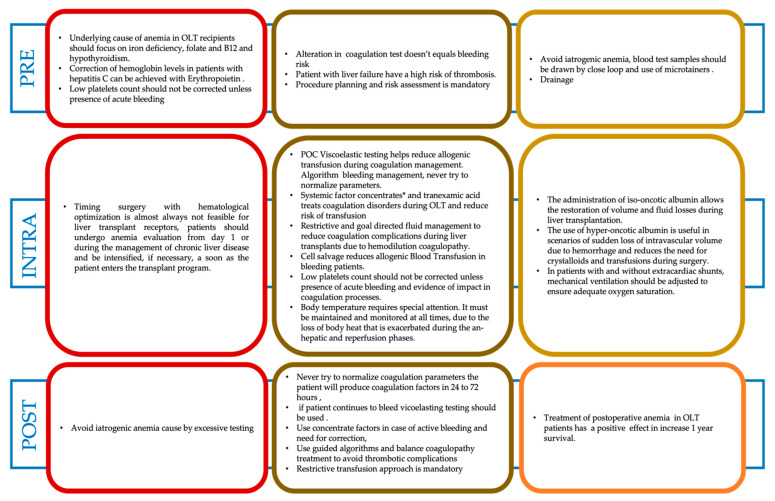
The three-pillar nine-field matrix of patient blood management in liver transplant recipients.

Another prevalent problem is post-liver transplant anemia (PLTA) [60]. One-third of patients with anemia had laboratory characteristics of iron deficiency anemia, 25% of patients of vitamin B12 or folic acid deficiencies, and 12.5% of patients’ hypothyroidism was evident. Early PLTA was an independent risk factor (OR 3.838) for mortality or graft failure in multivariate analysis [60]. The association between early PLTA and mortality or graft failure has been established despite a relatively short interval of three years of follow-up, which led to a limited number of events, accentuating the strength [60]. Therefore, a post-transplant anemia workup may be warranted in many of these patients. Whether treatment interventions, such as iron supplementation, in patients with early PLTA may prevent mortality or graft failure remains to be established in a prospective randomized controlled study. Patients with anemia also had a higher rate of malignancies at one-year post-liver transplantation after the competing risk analysis. Mechanisms attributed to anemia in patients with cancer also include inflammation, infection, nutritional deficits, blood loss, and bone marrow failure [61].

Liver transplant recipients have a high incidence of anemia prior to transplantation. Although anemia was corrected in half of the patients two years post-transplant, PLTA was still prevalent, with iron deficiency and CKD being important etiologies. Anemia is also closely associated with mortality or graft failure at the years of follow-up.

#### 3.4.2. Bleeding and Coagulation Management

Managing coagulopathy during the perioperative period in patients undergoing liver transplantation is a significant challenge. The limitations of conventional laboratory tests have made POC devices the standard modality for monitoring perioperative coagulation parameters [62,63,64,65]. Multiple studies on liver transplantation have shown the efficacy of POC coagulation management in reducing perioperative blood loss and the rate of blood product transfusion [29,42]. The risk of bleeding and transfusion is determined by the preoperative hemoglobin value, donor characteristics, surgical technique, antifibrinolytic therapy, prolonged duration of surgery, and the extensive trauma associated with it.

### 3.5. Coagulation Issues during Liver Transplantation

OLT has a dynamic equilibrium between the decreased procoagulant and anticoagulant levels, resulting in a lower hemostatic reserve. OLT is divided into the following surgical phases: pre-anhepatic, anhepatic, post-reperfusion, and postoperative [66].

#### 3.5.1. Pre-Anhepatic Phase

During this phase, surgical bleeding is the hallmark and is explained by increased portal hypertension with venous distension, which is difficult to control during hepatectomy. The dissection of adhesions and transection of collaterals for mobilization of the liver results in extensive surgical trauma and bleeding, with a negative contribution from the baseline easily alterable coagulopathy and fibrinolysis due to underlying liver disease [67]. In addition, restoring sudden volume loss by crystalloids may contribute to hemodilution and the perpetuation of coagulopathy [42].

#### 3.5.2. Anhepatic Phase

This phase extends from the occlusion of the hepatic vasculature to the reperfusion of the liver graft. As the hepatic veins/vena cava inferior are clamped, surgical blood loss is increased due to portal hypertension and liver outflow restriction. The absence of coagulation factor synthesis and clearance of activated fibrinolytic factors in the liver characterizes this phase. In addition, increased hypothermia due to the implantation of the cold-preserved liver graft results in further coagulopathy and platelet dysfunction. Increased levels of tPA secondary to its release from endothelial cells and decreased clearance in the absence of liver perfusion can lead to hyperfibrinolysis and severe bleeding [67,68].

#### 3.5.3. Post Reperfusion Phase

This phase is characterized by profound coagulation abnormalities related primarily to thrombocytopenia and a heparin-like effect on the donor’s liver [69]. Entrapment of platelets in a donor’s liver sinusoids can be profound enough to create a 50% gradient in platelet counts between arterial and venous circulation [16,29,30,58]. The ischemic donor liver endothelium releases heparinoids, which, on reperfusion, add to the residual heparin remaining as a part of the heparinization of the donor’s liver performed before harvest. These events can lead to uncontrolled diffuse bleeding within minutes of reperfusion in some patients [35]. Increased clearance of tPA and increased production of PAI-1 lead to a gradual resolution of hyperfibrinolysis.

#### 3.5.4. Postoperative Phase

This phase is characterized by thrombocytopenia and hypercoagulability. Platelet consumption and activation within the new liver destroy platelets, although the numbers return to normal by 14 days due to increased thrombopoietin production by the new liver. Hypercoagulability is a concern in most patients. Early recovery in procoagulants and elevated FVIII levels can exaggerate coagulability, especially when clubbed with delayed recovery in the production of anticoagulants (antithrombin, proteins C and S) [35].

### 3.6. Viscoelastic Testing for Coagulation Management in Liver Transplant

The thromboelastometry algorithm (Figure 2, Table 2 and Table 3) starts with the clinical detection of diffuse bleeding. This algorithm is an evidence-based management approach to aid treatment for fibrinolysis, clot firmness, and thrombin generation. Management of fibrinolysis and endogenous heparin-like effects is essential [36,62]. Standard coagulation tests in patients with cirrhosis are often characterized by thrombocytopenia and an increased international normalized ratio (INR). These abnormal results were not associated with an increased risk of bleeding due to a balance of hemostasis in this patient population [70,71]. As this rebalance occurs at a low level, it can be easily disturbed, resulting in bleeding and thrombosis [62,72].

Thromboelastometry parameters, EXTEM (A5_EX_ < 25 mm) and FIBTEM (CT_FIB_ > 600 s) are predictors of fibrinolysis [61,62], with FIBTEM being the most sensitive assay for fibrinolysis because the diagnosis is not affected by platelet-mediated clot retraction [73].

An A5_EX_ cut-off value < 25 mm (A10_EX_ of 35 mm and MCF_EX_ of 45 mm) and an A5_FIB_ cut-off value <8 mm (A10_FIB_ of 9 mm and MCF_FIB_ of 10 mm) are suitable for fibrinogen substitution in bleeding patients undergoing liver transplantation [74]. Lower levels of clot firmness compared with other settings have been reported to reduce the risk of thrombosis without increasing the risk of bleeding [64]. FIBTEM is superior to plasma fibrinogen concentration for predicting bleeding in liver transplantation than plasma fibrinogen concentration [65,66,67,68]. FIBTEM-guided algorithms for bleeding management during liver transplantation to guide fibrinogen substitution significantly reduce the transfusion requirements of blood products [74,75,76,77]. Preemptive administration should not be performed because FC did not reduce transfusion requirements in an RCT on liver transplantation [78].

Platelet transfusion guided by ROTEM reduces platelet transfusion by 64 to 75% without any additional bleeding events compared with transfusion guided by laboratory values (a platelet count < 50 × 10^9^/L) [76,77].

INR has a low value for assessing thrombin generation and bleeding risk in patients with cirrhosis [34]. Using the CT_EX_ cut-off value of 75s is superior for predicting bleeding. CT_EX_ guidance reduces FFP transfusion and PCC administration [75,76,77], helping to avoid the overtreatment of thromboembolic events [79,80].

Endogenous heparinization or heparin-like effect (HLE) is a coagulation feature well-described in patients undergoing liver transplantation [76,77]. A mild to severe (CT_IN_/CT_HEP_-ratio ≥ 1.25 to ≥ 2.0, respectively) HLE is detected in approximately 50% of cases after liver graft reperfusion (CT_IN_, 270–3312 s). The CT_IN_/CT_HEP_ ratio is more sensitive than APTT for identifying HLE. This phenomenon is associated with increased transfusion requirements. Detection of HLE during the anhepatic phase increases the 3-month mortality [80,81]. HLE is often self-limiting after hemodynamic stabilization during reperfusion [82,83,84]. If self-limitation does not occur, it can be reversed using small amounts of protamine [62,85,86,87,88].

Thromboelastometry can assess the risk of thrombosis, and ROTEM-guided bleeding management helps to avoid thromboembolic complications [85,86,87,88]. Hincker et al. [89] reported that preoperative APTT, INR, and platelet count could not predict postoperative thromboembolic events after major noncardiac surgery. INTEM and EXTEM A10 (A10_EX_ cut-off, >61.5 mm) were predictors of thromboembolic complications, contrary to FIBTEM, which does not predict thromboembolic events in the study. Moreover, different studies in patients with cirrhosis or liver transplantation demonstrated the predictive value of increased MCF_FIB_ (18 and 25 mm; risk ratio [RR] up to 4.8) for portal vein and hepatic artery thrombosis in patients with hereditary or acquired thrombophilia (e.g., antithrombin, protein C or protein S deficiency, factor V Leiden mutation, lupus anticoagulant, and antiphospholipid antibodies) and patients with hepatocellular or cholangiocellular carcinoma [90,91,92,93]. It is essential to emphasize the need for a correct dose adjustment and avoiding overtreatment with fibrinogen to protect the graft. Figure 3 shows the recommended timing for the ROTEM analysis in the OLT.

Thromboelastography (TEG) has also been used in these settings [94]. The same principle applies to obtain the best results from this device. Multiple studies have been conducted in the perioperative period of a liver transplant. The main findings of the published studies using TEG are coagulation alterations and correlation with INR values when compared with the TEG R line [94]. Increased PT and INR/PT values in cirrhotic patients often occur against a normal or near normal aPTT. Evaluating PT/INR in isolation does not consider thrombocytopenia and platelet dysfunction, as discussed earlier. TEG/ROTEM integrates these test results [86,87]. In patients with liver failure, standard coagulation tests do not correlate with the results of the viscoelastic test when using TEG. TEG determines an accurate coagulation profile that correlates with in vivo clinical presentation. However, the capacity to separate the effect of thrombocytopenia and hypofibrinogenemia in the treatment algorithm makes a better performance of ROTEM when detecting hyperfibrinolysis and factor X deficiency [87]. With this in mind, TEG may not help decision-making regarding cryoprecipitate or fibrinogen concentrate and platelet transfusion in liver transplants.

## 4. Hemostatic Drugs

### 4.1. Antifibrinolytics

While performing a LT, systemic fibrinolysis is not a significant predictor of mortality and is often transient in the anhepatic and reperfusion phases [95]. In the late anhepatic stage and postreperfusion period, fibrinolysis is caused by tPA being released from vessels but not being cleared by the liver. Clinical evidence for the use of antifibrinolytics during LT supports this practice [95,96].

Aprotinin was the first antifibrinolytic agent used during OLT [97]. There is a strong association between increased transfusion due to intraoperative fibrinolysis and bleeding during OLT [94]. The clinical use of aprotinin was suspended in 2007 after serious safety concerns in high-risk cardiac surgery patients were raised [97,98,99]. Suspension was lifted in Canada and Europe after the health authorities concluded that the respective studies suffered from significant flaws.

However, tranexamic acid (TXA) has become the most used antifibrinolytic agent worldwide. A meta-analysis [100] demonstrated that TXA infusion (10 mg/kg/h) reduces RBC and FFP transfusions compared with placebo. Additionally, the prophylactic use of TXA (bolus: 30 mg/kg plus infusion at 16 mg/kg/h) has been documented. Nowadays, antifibrinolytic therapy is restrictively used due to the potential risk of arteria hepatic thrombosis in most liver transplant centers [34]. EACA-treated patients have received more significant amounts of RBC, plasma, platelets, and cryoprecipitate with repeated bolus injections or prolonged infusion (EACA, from 5–10 to >10 g). The results did not show a difference in thromboembolic events between the EACA and non-EACA groups. Badenoch et al. [101], in patients treated with TXA (*n* = 367), reported a decrease in RBC and plasma requirements versus matched non-treated cohorts without increased HAT, portal vein, and other venous thromboses. The HALT-IT RCT showed an increased risk of thromboembolic events in patients with gastrointestinal bleeding treated with TXA [102].

Potentially fatal complications in OLT are intracardiac thrombosis (ICT) and pulmonary thromboembolism (PE), occurring in 0.36–4.0% of cases. Intraoperative mortality rates are as high as 30–55%, and overall mortality is as high as 45–68%. Antifibrinolytic therapy and ICT/PE association have not been established; however, TXA or EACA can hinder attempts at clotting dissolution by intravenously administered tPA [103,104]. ICT/PE occurs within 30 min of the reperfusion phase [105,106]. Transesophageal echocardiogram can be used for early diagnosis, and antifibrinolytics should be withheld in high-risk OLT cases during this phase.

### 4.2. Fibrinogen Concentrates

The use of fibrinogen concentrate (FC) overcomes the limitations of cryoprecipitates. Furthermore, it avoids an increase in factor VIII and von Willebrand factor, a prothrombotic factor in patients with cirrhosis [104,105,106,107]. All products are indicated to actively treat bleeding in patients with congenital fibrinogen deficiency. Stolt [103] reported on in vitro samples with fibrinogen concentrate supplementation the effectiveness of correcting coagulation testing resulting in higher plasma fibrinogen concentrations and improved clot strength. However, the selected dose for restoring clot strength in cardiac surgery patients differed among the three groups, and these results have implications for the choice of concentration and dosing [104].

FC has been shown to increase fibrin polymerization in viscoelastic testing with less variability than cryoprecipitate. This results in an increase in fibrinogen levels in plasma in a more predictable manner [103,107]. The manufacturing of FC mitigates pathogen transmissions [108] and minimizes allergic and other transfusion-related reactions [109].

During any hemostatic intervention, the risk of thromboembolic complications must be considered. HAT is the most feared event in the OLT population, with an incidence of 3–9% [110], a re-transplantation rate prevalence of 53%, and a mortality rate between 27% and 58% [111,112]. A review of 634 consecutive patients found that cytomegalovirus (CMV) infection and accessory hepatic artery reconstruction were significant predictors of HAT [109]. Evidence also showed that the patient’s age, indication for OLT, cold ischemic time, surgical time, and blood transfusion volume were not risk factors for HAT [110]. The incidence of HAT was 4.5% in OLT patients receiving FC and PCC compared with 3.6% in those who did not receive these concentrates [113]. Fibrinogen administration to correct hypofibrinogenemia has a positive effect on surgical bleeding. Fibrinogen concentrates effects on increasing plasma fibrinogen by 0.5 g/L has not been determined in OLT patients, and the administration should be adjusted to replace plasma fibrinogen levels in the range of expected values; it is highly recommended that it be guided by thromboelastometry (Table 3) [62].

### 4.3. Prothrombin Complex Concentrate

PCC has a higher concentration of factors than FFP (a difference of 0.8 to 1.2 IU/mL), allowing a rapid recovery of vitamin K-dependent factors in warfarin-treated patients without circulatory overload [113,114]. ESLD has reduced production of FII, FV, FVII, FIX, and FX [115,116,117]; four-factor PCC may help restore deficient factors, except for FV [117].

In contrast to modern four-factor PCCs containing significant amounts of proteins C and S, rFVIIa does not contain anticoagulants and is associated with increased thromboembolic events, particularly arterial thrombosis in liver transplantation, and should therefore be avoided [118]. In patients with severe liver damage due to active bleeding or invasive procedures, similar recoveries (1.3–1.4 IU/kg) have been observed among FII, FIX, and FX. The median dose of PCC in this case series was 25.7 IU/kg (1500 [1000–4000] IU). No adverse events, including thrombosis, were observed [115]. A dose of 25 IU/kg of PCC was used if EXTEM-CT above 80 seconds for active hemorrhage in 266 LT cases after restoring fibrinogen levels. PCC was used in 34.9% (*n* = 93) of the patients compared with 14.7% (*n* = 39) [116].

### 4.4. Thrombopoietin Receptors Agonist (TPO)

TPO has been approved in chronic liver disease (CLD) patients programmed for invasive procedures. The published studies demonstrated a reduction in the need for platelet transfusions by enhancing the TPO receptor activation, increasing megakaryocyte progenitor proliferation, and ultimately increasing platelet production [119,120]. The effects of TPO agonist have not been studied in patients programmed for a liver transplant, and the main controversy is the risk of thromboembolic events. The safety profile for OLT has yet to be established. There is no recommendation for their use in this setting.

## 5. Pediatric Considerations

Sufficient differences between pediatric and adult patients needing OLT require independent yet complementary documents. Children have distinct diseases, clinical susceptibilities, and physiological responses that distinguish them from those of adults. Significant differences among newborns, infants, children, and adolescents have been found [121]. A multidisciplinary pediatric OLT evaluation team should be skilled in pediatric conditions and adequately communicate LT’s processes, risks, and benefits to the family and the child. Indications for LT in the pediatric population include biliary atresia, metabolic/genetic conditions, acute liver failure, cirrhosis, liver tumor, immune-mediated liver, biliary injury, and other conditions. Acute liver failure (ALF) or acute decompensation may rapidly progress to death or irreversible neurological damage [122].

Despite coagulopathy for liver disease, pediatric patients also acquire coagulopathy due to the dilution of clotting factors. This complication depends on the type of fluid therapy and volume of transfused RBC. Anesthetic challenges during this stage include the maintenance of hemodynamic stability by adequate fluid and blood product administration and correcting coagulation abnormalities. Acquired fibrinogen deficiency should be treated with the substitution of cryoprecipitates or the intraoperative administration of FC (50 mg/kg) to treat hypofibrinogenemia (ROTEM FIBTEM maximum clot firmness < 7 mm) during major pediatric surgery. Substitution with other coagulation factors (FXIII levels > 60%) should be indicated with laboratory or viscoelastic testing results. FFP might help treat severe hemorrhage as an adjunct to primary treatment with coagulation factors. However, it has been shown that FFP does not adequately increase plasma fibrinogen concentration. A 20–30 mL/kg volume dose should be administered to increase fibrinogen concentration. Moreover, it should only be considered after excluding other factors influencing hemostasis (fibrinogen deficiency, hyperfibrinolysis, acidosis, hypothermia, and low platelet count or impaired platelet function).

Cardiac surgery results showed that fibrinogen concentrate is as efficient and safe as cryoprecipitate. Unfortunately, no study has assessed the response to fibrinogen concentration after FFP administration in children with liver transplants [122].

Thrombocytopenia, secondary to hypersplenism, has a high prevalence and impairs coagulation. Platelet administration must always be weighed against the risk of hepatic artery thrombosis in the new graft in the pediatric population [123].

Perioperative treatment with an antifibrinolytic agent reduces bleeding and transfusion requirements in pediatric LT and hepatectomy, with high bleeding risk [124]. Bleeding prevention should be considered a patient-based multimodal approach. Antifibrinolytic agents may play a central role in perioperative bleeding prophylaxis in the pediatric population [125].

Tranexamic acid should be administrated at a loading dose of 10 mg/kg over 15 min, followed by a 5 mg/kg maintenance infusion, to maintain adequate plasma concentrations. Tranexamic acid is necessary for all children undergoing surgery with a significant bleeding risk to treat fibrinolysis but not for routine prophylaxis. Post-reperfusion fibrinolysis is common in marginal grafts (e.g., donation after cardiac death) and should be considered in patients with cirrhosis undergoing liver resection [122,126].

The reference ranges for the ROTEM parameters in children are age dependent. Children aged 0–3 months exhibited accelerated coagulation and firm clot firmness despite showing prolonged standard plasma coagulation test results. In addition, platelet count and FXIII contribute to clot firmness in children’s fibrinogen concentration, as measured with the ROTEM assay. In addition, children aged 4–24 months showed 2.5% percentiles for clot strength, indicating a low reserve when exposed to hemodilution and blood [125,126,127]. Viscoelastic tests during pediatric LT are well-established. The main objective of perioperative coagulation management is to prevent severe coagulopathy and avoid thromboembolic complications and graft thrombosis in pediatric OLT.

In 2008, the NHS Quality Improvement Scotland released a report based on health technology assessment that integrated evidence such as clinical effectiveness, cost and benefits, organizational aspects, and more about patients. Viscoelastic testing is recommended instead of standard coagulation tests during cardiac surgery, and (National Institute for Health and Care Excellence, UK) guidelines recommend viscoelastic tests to monitor coagulation during and after surgery. It is associated with lower mortality risks, reduced complications, and lower transfusion rates and hospitalization time [127,128].

## 6. Blood Conservation Technics in Liver Transplantation

Several blood conservation strategies should be used for live-donor liver transplantation (LDLT). Postoperative blood conservation involves limited blood sampling and pediatric drainage tubes. Intraoperative techniques include Acute Normovolemic Hemodilution (ANH), cell salvage, and specialized surgical techniques. Combining preoperative blood augmentation with ANH can be especially effective in OLT, allowing the removal of greater quantities of whole blood without causing significant perioperative anemia. Surgical techniques to control portal hypertension and blood loss during OLT in recipients are also essential for blood conservation. Hepatic congestion is unique to OLT and is associated with impaired function of the transplanted lobe. With refining surgical techniques, blood transfusions in OLT have been significantly reduced. Strategies that aim to minimize blood loss and transfusion should be developed. Intraoperative blood salvage autotransfusion (IBSA) reduces the need for allogeneic blood transfusions [129]. A study of 150 consecutive OLT patients showed that IBSA reduced the need for blood transfusion [130]. The risk of bacterial infection in the operative field seems plausible; however, this has not been demonstrated [131].

Cell salvage is globally used in liver transplants as a blood conservation technique. A recent systematic review reported that no data were found indicating whether cell salvage usage resulted in a reduction of transfusion requirements. The use of cell salvage did not increase HCC recurrence and did not affect mortality. Moreover, they found no data for other measures, including perioperative complications. All studies included in this systematic review were observational and had a small sample size; despite the insufficient evidence to state a recommendation, the authors’ conclusion favors its use to reduce blood transfusion in this setting [65].

## 7. Patient Blood Management Consideration for Acute Liver Failure

In patients with acute liver failure (ALF), the principles of pillar one from the PBM program should be focused on avoiding iatrogenic blood loss; anemia treatment to normalize laboratory parameters is often tricky secondary to the acute illness and timing of presentation, and based on the current evidence, the hemostasis in ALF indicates a “rebalanced” state [132]. Prophylactic transfusion of blood products is undesirable and unwarranted [132]. Without a clear benefit, it may expose patients to complications such as volume overload and transfusion reaction, immunomodulation, and graft rejection [132]. Viscoelastic tests have also gained more recognition as potential tools in evaluating coagulopathy in patients with liver disease [132].

The recommendations for treating coagulation abnormalities and blood loss in patients with CLD have been ported to patients with ALF. The treatment should be coagulation support in the evidence of bleeding with the same objectives as patients with CLD. However, specific research for subgroups of patients with ALF and ACLF must be conducted to provide more personalized and precise treatments.

## 8. Conclusions

PBM program equates patient safety according to recent evidence in patients with liver transplants, and its principles have been endorsed by the WHO and are now considered standards of care. These principles should be rigorously applied to patients awaiting and undergoing OLT to improve outcomes and continue in the postoperative period.

## Figures and Tables

**Figure 2 biomedicines-11-01093-f002:**
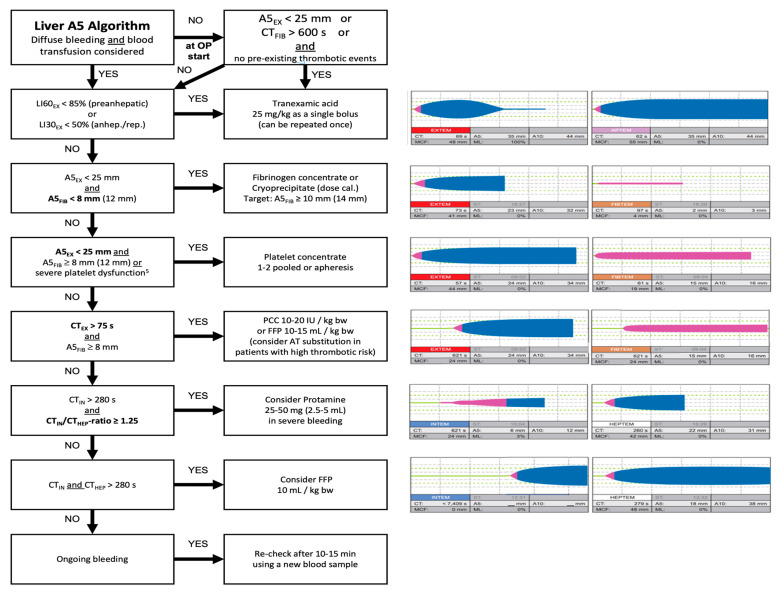
A5_EX_: amplitude of clot firmness 5 min after coagulation time in EXTEM, CT_FIB_: coagulation time in FIBTEM (CT_FIB_ > 600 s reflects a flat-line in FIBTEM). ML: maximum lysis (within one h run time), ACT: activated clotting time, CT_IN_: coagulation time in INTEM, CT_HEP_: coagulation time in HEPTEM, BW: body weight, A5_FIB_: amplitude of clot firmness 5 min after CT in FIBTEM, CT_EX_: coagulation time in EXTEM, PCC: prothrombin complex concentrate, FFP: fresh frozen plasma. LI60: Lysis Index (residual clot firmness in % of MCF) 60 min after CT, LI30: Lysis Index (residual clot firmness in % of MCF) 30 min after CT, IU: international units. AT: antithrombin, Cai2+: ionized calcium concentration, EACA: epsilon-aminocaproic acid, TXA: tranexamic acid, rFVIIa: activated recombinant factor VII. (Courtesy of Klaus Görlinger) Germany [62].

**Figure 3 biomedicines-11-01093-f003:**
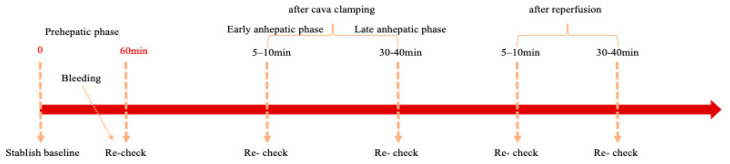
Timing of ROTEM analysis during orthotopic liver transplantation (OLT).

**Table 2 biomedicines-11-01093-t002:** Evidence-based algorithms for ROTEM (A5)-guided bleeding management in adult liver transplantation.

Step		Explanation
1	Check basic conditions	Temp. > 35 °C; pH > 7.3; Cai 2+ > 1 mmol/L; Hb ≥ 7 g/dL.
2	Antifibrinolytic therapy	FIB_CT_ > 600 s represents a flat line in FIBTEM. Pre-anhepatic hyperfibrinolysis increases mortality in OLT [62]; hyperfibrinolysis at/after reperfusion without diffuse bleeding may be self-limiting; if ML is less than 15%, consider avoidance of TXA treatment.
3	Fibrinogen dose calculation	Fibrinogen dose (g) = targeted increase in A5FIB (mm) × body weight (kg)/160. The correction factor (140–160 mm kg/g) depends on the plasma volume; 10 U Cryoprecipitate ≈2 g fibrinogen concentrate.
4	Platelet concentrate transfusion	Platelet transfusion is associated with increased mortality in liver transplantation.
5	Antithrombin (AT) substitution	Consider AT substitution in patients with an increased risk of thrombosis (e.g., primary biliary cirrhosis, Budd–Chiari syndrome, portal vein thrombosis, malignancies) and known pre-existing severe AT Deficiency.
6	Protamine	Endogenous heparin effect after liver graft reperfusion is self-limiting and does not require reversal by protamine. Consider protamine administration in severe bleeding.
7	Simultaneous interventions	Only in severe bleeding, consider a maximum of three interventions simultaneously. In moderate bleeding, consider a maximum of two interventions simultaneously. There was only one intervention simultaneously (in second or later analysis and mild to moderate bleeding).

OLT, orthotopic liver transplantation; Hb, hemoglobin; FIB, Fibtem; AT, antithrombin; U, units; mm, millimeter; kg, kilogram; g, gram.

**Table 3 biomedicines-11-01093-t003:** Evidence-based parameters for ROTEM A5.Pediatric liver algorithm.

Step		Explanation
1	Check basic conditions	Temp. > 35 °C; pH > 7.3; Cai 2+ > 1 mmol/L; Hb ≥ 7 g/dL.
2	Antifibrinolytic therapy	Tranexamic Acid pediatric dose is 15 mg/kg as a single bolus.
3	Fibrinogen dose calculation	Fibrinogen dose (g) = targeted increase in A5FIB (mm) × body weight (kg)/160. The correction factor (140–160 mm kg/g) depends on the plasma volume; 10 U Cryoprecipitate ≈ 2 g fibrinogen concentrate.
4	Platelet concentrate transfusion	Platelet concentrate 5–10 mL/kg single donor or apheresis.
5	Antithrombin (AT) substitution	Consider AT substitution in patients with an increased risk of thrombosis (e.g., primary biliary cirrhosis, Budd–Chiari syndrome, portal vein thrombosis, malignancies) and known pre-existing severe AT deficiency.
6	Protamine	Consider protamine (0.3–0.5 mg/kg) in severe bleeding.
7	Simultaneous interventions	Only in severe bleeding, consider a maximum of three interventions simultaneously. Moderate bleeding, consider a maximum of two interventions simultaneously. There was only one intervention simultaneously (in second or later analysis and mild to moderate bleeding).

All general parameters in ROTEM analysis are the same for adults and infants/children in liver transplantation algorithms. However, dosing needs to be weight-precise and adjusted. (Courtesy of Klaus Görlinger, Germany) [62]. OLT, orthotopic liver transplantation; Hb, hemoglobin; FIB, Fibtem; AT, antithrombin; U, units; mm, millimeter kg, kilogram; g, gram.

## Data Availability

Data sharing not applicable.

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
