# Peer review of "Patient Blood Management in Liver Transplant—A Concise Review"

_biomedicines, 2023, doi:10.3390/biomedicines11041093_

Round 1
Reviewer 1 Report (Previous Reviewer 3)
The authors have performed a very nice review about patient blood managment in LT.
The topic is of high interest and the manuscript is of very good quality. It should be very useful in this field.
I recommend the mansucripr for publication, but I advise minor chages, namely:
Table one - there is FPP? shoudn't be FFP (fresh frozen plasma?) either FPP or FFP there is no reference to the abreviation in the table
Figure 1 is very interestin, but leter size in the middle box od the "intra" section is too small. it should be a little bigger
In table 2 there are no references to the abbreviations. they shoud be in extent in the caption: AT, OLT, FIBTEM, etc
In line 185, it appears "Liver" with capital L
At the endo of the introduction section there should be a paragraph stating how was the research conducted, according to the PRISMA guidelines; where were searched the articles? time period, etc
Author Response
Please see the attachment

Reviewer 2 Report (Previous Reviewer 1)
I would like to congratulate the authors of the manuscript entitled “Patient Blood Management in liver transplant a concise review” for conducting a very well-documented and interesting literature review that sheds light upon a liver transplant related topic, regarding the contribution of a revolutionary concept known as the “patient blood management” in improving the outcome. This approach is based on the management and preservation of the patient's own blood while promoting patient safety and empowerment.
The manuscript’s topic is considered of actuality and interest, the paper presents in a clear manner with the help of several flow charts the topic in question.
The review is well structured and very coherent, thus leading to a high quality of presentation.
The conclusions are pertinent and well supported by the content of the manuscript. The cited references are considered up-to-date, and relevant to the research in question.
All the previously suggested content related comments were addressed upon this second manuscript submission. However, there are some minor form issues that need addressing:
· Throughout the text there are some words highlighted in red.
· “Conclusion” section needs numbering with 7.
· Section 6 title needs modifying to lowercase.
· Right column of Figure 1 needs orange colouring
· Row 446, subsection “Pediatric considerations” needs proper numbering.
Author Response
Please see the attachment

Reviewer 3 Report (New Reviewer)
This article by Pérez-Calatayud et al. is well written and well argued. Maybe a little long. I only have small corrections to improve the article somewhat. There are some errors in spaces before equal signs. There is also an error in the subtitles, eg, 3.4.1.2 figure but no text. There are also plenty of missing endpoints. Or also ";" which should be ".". There are also dots before parentheses that shouldn't be there.
Author Response
Please see the attachment

Reviewer 4 Report (New Reviewer)
Please, look at the file attached.

Round 2
Reviewer 4 Report (New Reviewer)
I really appreciated the revision the author made. Still some spelling corrections and punctuation are needed but a part from that the paper is really valuable to be published in the journal
Author Response
Please see the attachment

This manuscript is a resubmission of an earlier submission. The following is a list of the peer review reports and author responses from that submission.
Round 1
Reviewer 1 Report
I would like to congratulate the authors of the manuscript entitled “Patient Blood Management in adult and pediatric liver transplant - a concise review” for conducting a very well-documented and interesting literature review that sheds light upon a liver transplant related topic, regarding the contribution of a revolutionary concept known as the “patient blood management” in improving the outcome. This approach is based on the management and preservation of the patient's own blood while promoting patient safety and empowerment.
The manuscript’s topic is considered of actuality and interest, the paper presents in a clear manner with the help of several flow charts the topic in question.
The review is well structured and very coherent, thus leading to a high quality of presentation.
The conclusions are pertinent and well supported by the content of the manuscript. The cited references are considered up-to-date, and relevant to the research in question.
However, in order to make the manuscript easier to understand, and therefore more appealing to the reader, I suggest addressing the following minor issues:
I have 2 comments related to the content of the current manuscript:
- Consider revising the title, by eliminating the terms: adult and pediatric, because the information is redundant.
- Consider adding to the keywords: adult and pediatric.
- Consider adding comments to detail the utilility of the self-saver technique.
I have one comments related to the form:
- Figure 1 needs reconsidering in order to inprove readability.
Author Response
I would like to congratulate the authors of the manuscript entitled “Patient Blood Management in adult and pediatric liver transplant - a concise review” for conducting a very well-documented and interesting literature review that sheds light upon a liver transplant related topic, regarding the contribution of a revolutionary concept known as the “patient blood management” in improving the outcome. This approach is based on the management and preservation of the patient's own blood while promoting patient safety and empowerment.
The manuscript’s topic is considered of actuality and interest, the paper presents in a clear manner with the help of several flow charts the topic in question.
The review is well structured and very coherent, thus leading to a high quality of presentation.
The conclusions are pertinent and well supported by the content of the manuscript. The cited references are considered up-to-date, and relevant to the research in question.
Thanks to the reviewer for giving us the time to review the manuscript, the minor issues are address point by point.
However, in order to make the manuscript easier to understand, and therefore more appealing to the reader, I suggest addressing the following minor issues:
I have 2 comments related to the content of the current manuscript:
- Consider revising the title, by eliminating the terms: adult and pediatric, because the information is redundant.
We change the title and erased adult and pediatric words to avoid redundancy
- Consider adding to the keywords: adult and pediatric
We added the word adult and pediatric to the key points.
- Consider adding comments to detail the utility of the self-saver technique.
We added information of a recent systematic review and added the reference, unfortunately in this subject there are no published research of the benefits of its use. I hope with this addition we cover this issue.
I have one comments related to the form:
- Figure 1 needs reconsidering in order to improve readability.
We made modification to the figure to improve readability

Reviewer 2 Report
I read with interest this paper, which reviews the current evidence on blood products management in the liver transplant setting.
My comments
- I agree with a restrictive approach in patients with decompensated cirrhosis, and with the proposed strategies to correct anemia before transplantation. Nevertheless, this is often possibile only in stable patients, and not in those with ACLF or admitted to ICU for sepsis, hemorragic shock. This point should be discussed
- Figure 1 is quite difficult to read in the present format
- What about TPO agonists? Can they be an alternative to platelet transfusion?
- The Authors proposed a ROTEM-guided approach during liver transplantation to correct coagulopathy? Are there other approaches available? What about the use of TEG instead of ROTEM?
Author Response
Reviewer 2
I read with interest this paper, which reviews the current evidence on blood products management in the liver transplant setting.
Thank you for spending time in reviewing this manuscript
My comments
- I agree with a restrictive approach in patients with decompensated cirrhosis, and with the proposed strategies to correct anemia before transplantation. Nevertheless, this is often possible only in stable patients, and not in those with ACLF or admitted to ICU for sepsis, hemorrhagic shock. This point should be discussed
Patient blood management is a evidence based patient centered approach to improve outcomes by managing patient own blood and is not optimal blood use (which is also a part of this evidence in the program, while intended as a bundle of care the approach this has to be made case based, we understand that in some patients no every part of the program could be done however we discuss the available evidence of why it is important to try to deliver each part of the program. To try to discuss this approach in acute care there would be a lack of evidence to do so, however we can still deliver parts of the program (pilar 2 and 3) and still manage to impact outcomes in this type of patient’s ass expressed in lines 99 to 528.
- Figure 1 is quite difficult to read in the present format
We made modification to the figure to improve readability
- What about TPO agonists? Can they be an alternative to platelet transfusion?
in liver transplant settings some elective procedures are best delayed until after transplantation when platelet counts improve. No evidence of TPO agonist in liver transplant has been published for that reason is not included in the review
- The Authors proposed a ROTEM-guided approach during liver transplantation to correct coagulopathy? Are there other approaches available? What about the use of TEG instead of ROTEM?
We added information available on TEG for liver transplant

Reviewer 3 Report
Interesting manuscript, but in the tex there's no reference to pillar 3.
Also there are minor english mistakes such as in table 1 "Prolonged lend of stay"
Figure 1 is difficult to read.
Author Response
Reviewer 3
Interesting manuscript, but in the text there's no reference to pillar 3.
Thank you for taking the time to review our manuscript we added references for pilar 3
Also, there are minor English mistakes such as in table 1 "Prolonged lend of stay"
We reviewed and made corresponding language editing
Figure 1 is difficult to read.
We made modification to the figure to improve readability

Round 2
Reviewer 2 Report
In my opinion, the paper has not been improved after the first round of revision. Indeed, I have asked the Authors to discuss about the fact that restrictive, patient-centered approach about blood product transfusion is not always possible given peculiar features of cirrhotic patients and should be balanced with urgent events (e.g., ICU patient awaiting for LT with sepsis and anemia: do the Authors treat him/her with EPO?). This point has not been discussed.
Moreover, the section about TEG is very short, does not discuss about differences between TEG and ROTEM and is written with several typos.
Regards